# Dichloroacetate Radiosensitizes Hypoxic Breast Cancer Cells

**DOI:** 10.3390/ijms21249367

**Published:** 2020-12-09

**Authors:** Sven de Mey, Inès Dufait, Heng Jiang, Cyril Corbet, Hui Wang, Melissa Van De Gucht, Lisa Kerkhove, Ka Lun Law, Hugo Vandenplas, Thierry Gevaert, Olivier Feron, Mark De Ridder

**Affiliations:** 1Department of Radiotherapy, Universitair Ziekenhuis Brussel, Vrije Universiteit Brussel, 1090 Brussels, Belgium; Sven.de.mey@vub.be (S.d.M.); Ines.dufait@vub.be (I.D.); jiangheng1981@gmail.com (H.J.); hui.wang.ecnu@gmail.com (H.W.); Melissa.Van.De.Gucht@vub.be (M.V.D.G.); Lisa.Kerkhove@vub.be (L.K.); ka.lun.law@vub.be (K.L.L.); Thierry.Gevaert@uzbrussel.be (T.G.); 2Pole of Pharmacology and Therapeutics (FATH), Institut de Recherche Expérimentale et Clinique (IREC), UCLouvain, 1200 Brussels, Belgium; cyril.corbet@uclouvain.be (C.C.); olivier.feron@uclouvain.be (O.F.); 3Department of Medical Oncology, Universitair Ziekenhuis Brussel, Vrije Universiteit Brussel, 1090 Brussels, Belgium; Hugo.Vandenplas@vub.be

**Keywords:** dichloroacetate, hypoxic radiosensitivity, breast cancer, reactive oxygen species

## Abstract

Mitochondrial metabolism is an attractive target for cancer therapy. Reprogramming metabolic pathways can potentially sensitize tumors with limited treatment options, such as triple-negative breast cancer (TNBC), to chemo- and/or radiotherapy. Dichloroacetate (DCA) is a specific inhibitor of the pyruvate dehydrogenase kinase (PDK), which leads to enhanced reactive oxygen species (ROS) production. ROS are the primary effector molecules of radiation and an increase hereof will enhance the radioresponse. In this study, we evaluated the effects of DCA and radiotherapy on two TNBC cell lines, namely EMT6 and 4T1, under aerobic and hypoxic conditions. As expected, DCA treatment decreased phosphorylated pyruvate dehydrogenase (PDH) and lowered both extracellular acidification rate (ECAR) and lactate production. Remarkably, DCA treatment led to a significant increase in ROS production (up to 15-fold) in hypoxic cancer cells but not in aerobic cells. Consistently, DCA radiosensitized hypoxic tumor cells and 3D spheroids while leaving the intrinsic radiosensitivity of the tumor cells unchanged. Our results suggest that although described as an oxidative phosphorylation (OXPHOS)-promoting drug, DCA can also increase hypoxic radioresponses. This study therefore paves the way for the targeting of mitochondrial metabolism of hypoxic cancer cells, in particular to combat radioresistance.

## 1. Introduction

Breast cancer is the most common cancer in women globally and results annually in 627,000 deaths [1]. In the last decades, significant progress has been made in the treatment of breast cancer. However, only limited therapies are available for patients with triple-negative/basal-like breast cancers [2,3,4]. The standard of care for the treatment of high-risk breast cancers consists of neoadjuvant chemotherapy and surgery, followed by postoperative whole-breast/chest wall irradiation. Nowadays, researchers are focusing either on hypofractionation of the adjuvant radiotherapy (FAST-Forward trial [5]) or the combination of chemotherapy with preoperative radiotherapy. The preoperative radiotherapy approach could result in improved disease-free survival and quality of life [6,7,8,9,10,11].

The main effect of radiation, particularly for low linear energy transfer radiation, is the induction of reactive oxygen species (ROS). During radiotherapy, ROS are created by the radiolysis of water in extracellular environments, which are toxic to tumor cells and nearby normal tissue. About two-thirds of radiation-induced DNA damage are attributed to ROS in mammalian cells [12]. The response of the cells to radiation-induced DNA damage is strongly dependent upon the presence of oxygen. Oxygen molecules can indeed fix DNA damages produced by free radicals. This is called the “oxygen fixation hypothesis” [12,13]. In the absence of oxygen, DNA radicals are reduced by compounds containing sulfhydryl groups, which repair the DNA to its original form. Following this hypothesis, hypoxia, defined by low oxygen levels in the tumor, is one of the principal causes of clinical failure of radiotherapy [14,15]. Hypoxia is a common feature of the tumor microenvironment. ROS and hypoxia are two factors with opposite effects on the radioresponse of the tumor [16]. The generally accepted hypothesis stated that less oxidative stress occurred in hypoxic regions of the tumor due to the shortage of ROS substrate oxygen. However, recent evidence revealed that under hypoxic conditions, cells generate more ROS, mostly via mitochondrial metabolism [17,18,19,20].

A defining hallmark of tumor cells is the ability to alter their metabolism, providing them with energy and metabolites required for their growth and survival in nutrient- and oxygen-limiting conditions. Yet, in the presence of O_2_, cancer cells also adapt their metabolism toward glycolysis, diverting mitochondrial pyruvate oxidation to lactate production [21,22]. This effect is referred to as the Warburg effect. Recent reports indicate that the Warburg effect is implicated in the resistance to cytotoxic stress induced by either chemotherapy or radiotherapy [23,24,25,26,27]. In this way, treatment methods that block or reduce glycolytic metabolism may increase tumor cell sensitivity to radiotherapy.

Under hypoxic conditions, hypoxia-inducible factor 1-alpha (HIF1α) causes an increase in the expression of pyruvate dehydrogenase kinases (PDK1–4) [28]. These enzymes are responsible for the metabolism switch in the mitochondria by regulating the phosphorylation status (i.e., activity state) of pyruvate dehydrogenase (PDH), which is a major gatekeeper protein between glycolysis and mitochondrial oxidative phosphorylation (OXPHOS). Dichloroacetate (DCA), a small-molecule PDK inhibitor, can reverse the Warburg effect by activating PDH and redirecting pyruvate metabolism back into the mitochondria. Inhibition of PDK by DCA is used to treat lactic acidosis and hereditary mitochondrial diseases [29,30]. Altogether, these observations have led to consider DCA as a potential anticancer drug [30,31].

DCA has been demonstrated to enhance the radiosensitivity of colon and prostate cancer cells as well as esophageal squamous cell carcinoma and glioblastoma tumors [32,33,34,35]. Still, no similar research has been performed on breast cancer cells. The main mechanism of radiosensitization in these models was attributed to oxidative stress. At the same time, cell-cycle arrest at the G2-M phase and a reduced mitochondrial reserve capacity also contributed to the radiosensitizing effects. Based on the previously described results, two clinical trials (one in head and neck carcinoma and one in glioblastoma) investigating the antitumor effects of combined DCA with radiotherapy treatment are currently ongoing [36,37]. All preclinical research has been performed under aerobic conditions, but no data about the effects of DCA treatment under hypoxic conditions is available. Therefore, in the present study, we first examined the hypothesis that DCA lowers lactate and switches the metabolism from a glycolytic phenotype to OXPHOS. Next, we determined whether DCA can radiosensitize hypoxic breast cancer cells and further examined underlying mechanisms. The findings of this study may have important implications for clinical trials aimed at using PDK inhibitors to improve the radioresponse in breast cancer patients.

## 2. Results

### 2.1. High Expression of PDK1 and PDK3 in Triple-Negative Breast Cancer (TNBC) and Basal-Like Breast Cancer Patients is Correlated with a Hypoxia-Related Gene Signature

First of all, by using the online and publicly accessible cBioPortal for Cancer Genomics, we analyzed the mRNA levels of the four different PDK isomers, namely PDK1, PDK2, PDK3 and PDK4, in patient-derived data from the primary breast cancer tumor dataset TCGA (PanCancer Atlas and Cell 2015) [38,39]. We showed a significant gain (*p* < 0.0001) in the expression of PDK1 and PDK3 in TNBC versus non-TNBC and in basal-like breast cancers relative to other subtypes such as luminal A, luminal B and HER2-enriched breast cancers (Figure 1a–c). Notably, the upregulation of PDK1 and PDK3 in breast cancer patients could be correlated with higher Ragnum and Buffa hypoxic scores [40,41] (Figure 1d). These hypoxia scores are based on differential expression of specific hypoxia-related genes and can be freely accessed through cBioPortal for Cancer Genomics. The observed upregulation of PDK1 and PDK3 in basal-like breast cancers and their correlation with a hypoxic phenotype (which is linked to the activation of a HIF1α−dependent transcriptional program), suggests that the use of PDK inhibitors could serve as an attractive therapeutic modality for hypoxic radiosensitization purposes [42,43,44].

### 2.2. DCA Decreased Phosphorylated PDH, Extracellular Lactate Levels and Extracellular Acidification Rate (ECAR)

We started our in vitro experiments by performing viability assays (Appendix A) in order to determine the growth-inhibiting properties of DCA. DCA reduced cell viability in a dose-dependent manner in EMT6 and 4T1 cancer cells regardless of the O_2_ status (Appendix A). A proliferation assay also showed that increasing DCA concentrations led us to observe a shift from growth delay to cytostatic and even cytotoxic effects (Appendix A).

Next, we investigated the influence of DCA on PDK1–4 activity and the metabolism of TNBC cells. PDH is a key gatekeeper protein of glycolysis and mitochondrial OXPHOS, meaning that PDH catalyzes the rate-limiting decarboxylation of pyruvate into acetyl-CoA. PDH is inhibited through phosphorylation (on Ser293) by PDK, and this inhibition can be reversed via dephosphorylation by pyruvate dehydrogenase phosphatase (PDP) [45,46] (Figure 2a). At acceptable toxicity doses (30 mM, 45 mM and 60 mM), we evaluated the effect of DCA on PDK activity by measuring phosphorylated PDH (p-PDH) levels, lactate in the extracellular medium and the ECAR of the cells in real-time (Figure 2b–e). All three doses of DCA decreased the amount of p-PDH in EMT6 and 4T1 cells in a dose-dependent manner under both oxygenated and hypoxic conditions. The decrease in p-PDH was significant for all the doses of DCA in EMT6, whereas in 4T1, only 45 mM and 60 mM significantly decreased p-PDH in hypoxic conditions (Figure 2b,c). Evaluating the effect of lower doses of DCA, we found that in EMT6 cells the amount of p-PDH starts to decrease when treated with 3 mM DCA and in 4T1 cells when treated 10 mM DCA (Appendix A). The amount of lactate in the medium was elevated under hypoxic conditions in comparison with aerobic conditions in both cell lines, supporting a net increase in glycolytic turnover in O_2_-deprived cells [47,48]. In agreement with the Western blot results, treatment with DCA led to a dose-dependent decrease of lactate in the medium for both cell lines (Figure 2d). Although the reduction in lactate release was dramatic under aerobic conditions, a dose-dependent reduction in the production of the end-glycolytic product was also observed under hypoxia. Finally, DCA from a dose of 7.5 mM, caused a time-dependent reduction in ECAR in both EMT6 and 4T1 (Figure 2e). The initial drop in ECAR after treatment with doses lower than 30 mM DCA was offset after 2.5 h in EMT6. In 4T1, we observed the same effect with DCA doses lower than 15 mM. These results indicate that the treatment of murine TNBC cell lines with DCA inhibits PDH phosphorylation and reduces the extent of glycolysis both under aerobic and hypoxic conditions.

### 2.3. DCA Induces ROS Production in Hypoxic Cancer Cells

The inhibition of PDK and activation of PDH are reported to be associated with the upregulation of intracellular ROS [49,50]. ROS is of pivotal importance in generating DNA damage following radiation. We examined the ROS levels in EMT6 and 4T1 cells under aerobic and hypoxic conditions by using the CM-H_2_DCFDA probe. As shown in Figure 3a,b, DCA triggered a dose-dependent ROS production in both EMT6 and 4T1. Under aerobic conditions, only the highest dose (60 mM) of DCA generated a significant ROS upregulation up to five-fold compared to control for EMT6 cells. In 4T1 cells, no significant upregulation of ROS was detected under aerobic conditions. The upregulation of ROS was partly counteracted by the addition of the ROS scavenger N-acetyl-cysteine (NAC). Under hypoxic conditions, we demonstrated a dose-dependent increase in ROS reaching up to 15-fold in EMT6 cells and up to five-fold in 4T1 cells; the lowest DCA dose actually led to a significant increase in ROS on both cell types (Figure 3b).

We reasoned that the increased ROS production under hypoxia could result from the combined effects of altered mitochondrial electron transport chain (due to the reduction in O_2_ as the final electron acceptor) and the forced DCA-driven pyruvate oxidative metabolism. Using the Seahorse analyzer, we found that DCA had no impact on basal respiration and ATP production but significantly decreased maximal respiratory capacity in EMT6 and 4T1 tumor cells (Figure 3c). Another possibility is that DCA increases NAD(P)H levels, which is linked to increased production of ROS. We saw that under aerobic conditions, the treatment of DCA had a dose-depend increase of NAD(P)H in EMT6 and 4T1 cells (Figure 3d). Under hypoxic conditions, we demonstrated a possible increase in NAD(P)H in EMT6 cells that were treated with 60 mM DCA but no increase in 4T1 cells (Figure 3d). Together with the above ROS data, these findings suggest that while the mitochondrial metabolism is still preserved, a local increase in ROS production upon DCA exposure may alter the integrity and the function of mitochondria.

### 2.4. DCA Radiosensitizes Hypoxic Cells Mediated by ROS

First, we looked at the hypoxia-induced radioresistance of the cells. We found a severely impaired radioresponse comparing hypoxic with aerobic conditions, with an oxygen enhancement ratio of 2.7 and 2.3 for EMT6 and 4T1 tumor cells, respectively (Figure 4a). In this setting, we observed that DCA treatment caused a small effect of intrinsic radiosensitization in EMT6 but not in 4T1 cells (Figure 4b). Interestingly, in line with the results of ROS generation under hypoxic conditions, 60 mM DCA significantly (*p* < 0.05) overcame hypoxic radioresistance with enhancement ratios of 2.3 and 1.5 at 60 mM for EMT6 and 4T1 tumor cells, respectively (Figure 5a). The radiosensitizing effect was reversed by NAC in both EMT6 and 4T1 cells (Figure 5b). The primary cause of radiation-induced cell death by ROS is through the induction of double-strand breaks in the DNA (ds-DNA) [12,51]. We therefore examined the ds-DNA damage after treatment with DCA by quantifying the phosphorylation status of γH2AX under hypoxic conditions. As shown in Figure 5c, DCA increased the formation of ds-DNA damage in both EMT6 and 4T1 in a dose-dependent manner.

### 2.5. DCA Radiosensitizes 3D Cell Cultures (Spheroids)

The above findings led us to examine whether DCA could also improve the radiosensitivity of three-dimensional (3D) cell culture models (Figure 6a–c) that better mimic the physicochemical properties of the tumor microenvironment, including oxygen gradients. Using spheroids obtained from EMT6 and 4T1 cell cultures, we measured spheroid growth after DCA treatment and radiotherapy (Figure 6a). DCA treatment alone resulted in a small growth delay on the EMT6 cells but did not alter the growth of 4T1 spheroids (Figure 6b). An 8 Gy irradiation reduced the growth of tumor spheroids, an effect further accentuated by the combination with DCA treatment (Figure 6a,c). Of note, whereas cytotoxic effects were observed in 4T1 spheroids (as revealed by halos of dead cells), cytostatic effects were observed in EMT6 spheroids (Figure 6a).

### 2.6. The Combination of DCA and Radiotherapy Does Not Delay Tumor Growth In Vivo

We next examined whether the in vitro benefit of the DCA combination to radiotherapy could be validated in vivo (Appendix A). Mice injected with EMT6 or 4T1 breast cancer cells were exposed to either single fraction (12 Gy or 15 Gy, respectively) (Appendix A) or fractionated (5*4 Gy or 5*6 Gy, respectively) (Appendix A) radiation. The single and fractionated radiation doses per tumor type are similar in biological effective dose (BED) and differ in function of the intrinsic radiosensitivity of the used cell lines. Importantly, DCA injection either i.p. or i.t. for 10 days was safe without inducing noticeable toxicity (Appendix A). Radiation alone delayed tumor growth in EMT6 for seven days with a single fraction and four days for fractionated radiation (Appendix A). In 4T1 tumors, radiation delayed tumor growth with five days for single fraction and 10 days for fractionated radiation, as expected (Appendix A). DCA (300 mg/kg), through either intraperitoneal (ip) or intratumoral (it) injection, did not delay tumor growth and neither did the combination of DCA with radiation (Appendix A). Next, we validated whether the treatment of DCA could induce hypoxia in tumors (Appendix A). Although the extent of pimonidazole-stained hypoxia was not altered in EMT6 tumors, a trend toward a decrease in hypoxia was observed in response to DCA in 4T1 tumors.

## 3. Discussion

The purpose of this study was to examine whether DCA targeting of mitochondrial metabolism could sensitize TNBC/basal-like breast cancer cells to radiotherapy. Most TNBC and basal-like breast cancers are aggressive tumors for which treatment options are limited and the prognosis is poor [2,3,4]. These tumors exhibit an enhanced glycolytic phenotype that supports their poor prognosis and is additionally correlated with radioresistance [52]. In the current study, we found that mRNA levels for two of the four PDK isoforms (PDK1 and PDK3) are upregulated in the TNBC and basal-like breast cancer subtypes. The overexpression of PDKs has been detected in multiple human tumor samples [42,53,54,55,56,57,58,59,60,61], and many cancer cell lines have substantial upregulation of PDK isoforms [50,62,63]. It has been reported that PDK overexpression is associated with a poor prognosis in a variety of tumor types [53,54,55,56,57,58,59,60,61]. The overexpression of PDKs in cancer cells is influenced by several transcription factors, such as HIF1 [28,64]. HIF1 actively suppresses OXPHOS by transactivating the genes encoding PDK1 and PDK3 directly. PDK in turn phosphorylate and inactivate PDH [42]. As such, the upregulation of PDKs in cancer can be drawn instantly back to both transforming mutations and the hypoxic tumor microenvironment. Consistent with these findings, we observed that the upregulation of PDK1 and PDK3 mRNA is correlated with hypoxia-related gene profiles. Metabolic reprogramming upon targeting of the PDK enzymes, to switch from glycolysis to OXPHOS, therefore appears as a promising therapeutic avenue to treat breast cancers with limited therapy options.

The major finding of our study is that the PDK inhibitor DCA can lower the glycolytic activity of breast cancer cells in the presence of oxygen but also under hypoxia [29]. We found that DCA lowers the amount of phosphorylated PDH and triggers a dose-dependent decrease in extracellular lactate levels and ECAR in aerobic and hypoxic cells. We next combined DCA and radiotherapy under the hypothesis that by reversing the glycolytic phenotype and directing more pyruvate into mitochondrial oxidation, tumor cells could produce more ROS and become more sensitive to radiation. Intrinsic radiosensitization effects of DCA have actually been reported for glioblastoma cells [34,35], non-small cell lung carcinoma (NSCLC) cells [65,66], colorectal [35], prostate cancer cells [32] and radioresistant medulloblastoma cells [67]. The proposed mechanisms are cell cycle arrest in the G2-M phase, creating extra DNA damages and consecutive cell death in response to increased mitochondrial ROS production. In the current study, we found that whereas minimal radiosensitizing effects were observed with the highest nontoxic concentration of DCA under aerobic conditions, DCA strongly radiosensitized hypoxic breast cancer cells, both in 2D as 3D spheroids. Although in theory ROS are associated with oxidative mechanisms, a partnership exists between hypoxia and ROS in tumors. Hypoxia enhances ROS generation via prolongation of the lifetime of the semiquinone radicals; reciprocally, ROS assist tumor cells in adapting to hypoxia via stabilization of HIF1-α [16,68]. However, extracellular insults could break down this partnership via triggering excessive ROS production, which impairs mitochondrial respiration and thus decreases the hypoxic fraction in tumors [16,69,70]. In this context, arsenic trioxide inhibits the oxygen consumption of tumor cells through an increase of intracellular ROS, leading to enhanced radioresponse [71]. The suppression of glycolysis is also reported to increase the radioresponse. This can be done via ritonavir (glucose transporter inhibitor), 2-deoxyglucose (hexokinase inhibitor) and lonidamine (hexokinase inhibitor), which are under investigation in clinical trials in different types of cancer [72,73,74,75]. Another possibility is that ROS is created after treatment with DCA because of the induction of NADPH oxidase [76]. However, no direct evidence was found to conclude that DCA can upregulate NADPH oxidase [77]. We observed dose-dependent increase of NAD(P)H under aerobic conditions, but not under hypoxic conditions in EMT6 and 4T1 cells. We therefore hypothesize that the increase of NAD(P)H and upregulation of NADPH oxidases only plays a minor role in the increase of ROS under hypoxic conditions. In our hands, the primary mechanism of the radiosensitization effects observed is very likely to result from the many fold increases in ROS production (up to 15-fold) after DCA treatment under hypoxic conditions.

In line with the literature, we demonstrated the need for supraphysiological concentrations of DCA to elicit alterations to metabolic activity, increased ROS formation and radiosensitization [29]. Our central hypothesis is that these effects are the consequence of PDK inhibition produced by DCA [78]. However, the concentrations required to induce the measured results are several times higher than the inhibition constant (Ki) of PDK1–4. It is noteworthy that DCA exists physiologically as an anion, is relatively membrane-impermeable despite its small size and therefore requires the mitochondrial pyruvate carrier for mitochondrial uptake [79,80]. However, the conjugation of DCA to a lipophilic carrier enhanced mitochondrial transport. It reduced the IC50 value of DCA from millimolar to the low micromolar range, which is well within the Ki of PDK1–4 range [81]. DCA mimics the effective inhibition of PDK1–4 by siRNA, and DCA added to PDK siRNA did not have additional effects [49,50,60,82,83,84,85,86,87,88]. Next, a small-molecule-like DCA might directly or indirectly affect other cellular and molecular targets. Recent research found that DCA treatment increased the concentration of every TCA intermediate but did not affect glucose uptake or glycolysis [89]. Other researchers showed evidence suggesting DCA can increase de novo CoA biosynthesis. Since high concentrations of CoA can be toxic for cells, this metabolic effect could be partially responsible for cancer cell toxicity mediated by DCA [90]. Recent research introduced a novel hypothesis, suggesting that the efficiency of DCA against cancer may be derived from its ability to antagonize acetate. High acetate levels can enhance DNA, RNA and protein synthesis. Additionally, it can be associated with anticancer drug resistance [91]. Lastly, researchers found that DCA could activate the AMPK signaling pathway, leading to a cascade of downstream metabolic and anticancer effects [92,93]. Nonetheless, our hypothesis remains that under hypoxic conditions, pyruvate’s delivery into mitochondria causes an upregulation of ROS levels, which radiosensitizes cancer cells. We failed to recapitulate these effects in vivo, and more work is needed to determine how to translate the radiosensitizing effects of DCA in hypoxic tumor compartments. Pharmacokinetics issues related to DCA administration in vivo can be excluded, and the dose of DCA used (150 mg/kg) is well within the doses used in the literature [29]. The human equivalent dose of our DCA dose used in vivo is 12 mg/kg/d, well within the tolerable zone used in clinical trials. A possible explanation for our failed in vivo experiments is that a higher DCA dose is needed to have a radiosenstizing effect in vivo. In the last 30 years, DCA has indeed been administered with reasonable success as an investigational drug for the treatment of type 2 diabetes, acquired and congenital hyperlipoproteinemia, myocardial ischemia, acquired and congenital lactic acidosis and more recently cancer [29,30]. Several phase I/II trials are investigating DCA safety and its activity as an anticancer agent. DCA is rapidly absorbed and can even cross the blood–brain barrier. Two phase I trials examined the safety of oral DCA in patients with recurrent malignant brain tumors or metastases to the brain from noncentral nervous system cancers [94,95]. These studies indicated that DCA is generally well-tolerated in patients. An alternate explanation for the puzzling difference between in vitro and in vivo effects is actually more likely to be linked to changes in the metabolic phenotypes when cancer cells are (ectopically) injected in vivo. Indeed, increased in vitro radiosensitivity by DCA is achieved under conditions of hypoxia and high glycolytic metabolism so that the glycolysis-to-OXPHOS shift can be observed upon DCA treatment together with extra ROS upregulation.

The limited extent of hypoxia in in vivo tumors may thus reflect a restricted capacity of DCA to induce a shift that is already present within well-oxygenated tumors largely dependent on OXPHOS. Further investigations are warranted to test the radiosensitizing effects of DCA in mouse breast tumor models characterized by limited angiogenesis and elevated hypoxic fractions.

In conclusion, we demonstrate that DCA overcomes hypoxic radioresistance of breast cancer cells in 2D and 3D systems, which can primarily be attributed to the upregulation of ROS. Remarkably, the DCA-induced shift in glycolysis-to-OXPHOS metabolism also creates oxidative stress under hypoxic conditions. DCA has been used for many years to treat metabolic conditions and inherited mitochondrial diseases. In the last decade, DCA has been largely repurposed as an anticancer drug with promising preclinical data, case reports and clinical trials, as described before. The current preclinical results indicate that further investigation of the anticancer potential of DCA is necessary considering that hypoxic tumor cells are not spared by the PDK inhibitor that may induce lethal oxidative stress, in particular when combined with radiotherapy.

## 4. Materials and Methods

### 4.1. TCGA Breast Cancer Cohort Analysis

The PDK1–4 mRNA expression profiles (RNA Seq V2 RSEM or log RNA Seq V2 RSEM) were queried from the cBioPortal website in the form of *z*-score-transformed data [38,39]. The queried data were assessed from 1084 publicly available cases of breast cancer of the TCGA PanCancer Atlas and from 817 breast cancer cases of the TCGA Cell 2015 database. For the TCGA cell 2015 dataset, the analysis of PDK1–4 was performed by comparing the triple-negative subpopulation with the rest of the breast cancer cases. Triple-negative breast cancer was determined by a “negative” status for the immunohistochemistry scores of estrogen receptor (ER), progesterone receptor (PR) and human epidermal growth factor receptor 2 (HER2) genes (total of 83 cases). In the TCGA PanCancer Atlas database, the analysis of PDK1–4 expression was performed across different Pam50 categories. Pam50 is a 50-gene signature that classifies breast cancer into five molecular intrinsic subtypes: Luminal A, Luminal B, HER2-enriched, Basal-like and Normal-like. TCGA breast cancer mRNA expression and clinical data were analyzed directly on the cBioPortal website or downloaded for further analysis. The Buffa and Ragnum hypoxia score analysis of the samples where PDK1–4 have an expression of a *z*-score higher than 2 were done directly on the cBioPortal website.

### 4.2. Cell Lines and Chemicals

The murine mammary adenocarcinoma EMT6 cell line was kindly provided by Edith Lord (University of Rochester, Cancer Center, New York), and 4T1 cells were obtained from American Type Culture Collection. All experiments were performed in Roswell Park Memorial Institute 1640 medium (Thermo Fisher Scientific, Waltham, MA, USA) supplemented with 10% fetal bovine serum (Greiner Bio-One, Kremsmünster, Austria). HEPES buffer was used for all treatments of the cells. Chemicals were obtained from Sigma-Aldrich (Sigma-Aldrich, St. Louis, MO, USA) unless otherwise stated

### 4.3. Treatments

EMT6 and 4T1 were grown to confluence and treated with DCA for 16 h at indicated concentrations. N-acetyl cysteine (NAC) was added at 10 mM or 20 mM to cultures both 1 h prior to and during treatment with DCA. Afterward, cultures were used for further analysis as described below. Treatment was performed either in aerobic conditions or hypoxic conditions. Hypoxia was induced by incubation in nitrogen/carbon dioxide-balanced gas containing 1% oxygen [96].

### 4.4. MTT Assay

Cytotoxicity of DCA was assessed by MTT assay as described elsewhere [97,98]. Briefly, cells were grown in 96-well plates and treated with indicated concentrations. After treatment, the medium was aspirated, and 50 µL of the MTT reagent (5 mg/mL) was added for 1.5 h. Next, 200 µL of MTT solvent (19:1 DMSO/HCL) was added and admixed to dissolve the formazan crystals generated inside of the cells. Absorbance was measured at a wavelength of 540 nm by using a spectrophotometer (Bio-Rad Laboratories, Hercules, CA, USA). Cell viability was determined by normalizing the treated cells to the untreated control cells.

### 4.5. Kinetic Growth Assay

The influence of DCA on the proliferation of the EMT6 and 4T1 cells was assessed by using a kinetic growth assay. Cells were grown up to confluence in 96-well culture plates and treated with DCA at indicated concentrations in 6 replicates. Photomicrographs were taken every two hours using an Incucyte live cell imager (Essen Biosciences, Newark, United Kingdom), and the confluence of the cultures was measured using Incucyte software (Incucyte ZOOM 2018A, Essen Biosciences) over 80 h in culture.

### 4.6. Western Blot

Western blot analyses were performed as previously described [99]. Briefly, cells were lysed in 1% triton-X buffer supplemented with a phosphatase inhibitor (P5726), protease inhibitor (P8340) and leupeptin trifluoroacetate (L2023). Lysates were centrifuged, and protein concentration was determined using the Bio-Rad DC protein assay (Bio-Rad 500-0116). Equivalent amounts of protein were loaded on a 12% resolving acrylamide gel. Protein transfer was conducted overnight at 4 °C using a nitrocellulose membrane (0.45 µM, Thermo 88018, Thermo Fisher Scientific). The membranes were blocked with 5% BSA in TBS and washed (TBST). Blocked membranes were labeled with primary antibody overnight at 4 °C. Primary antibodies were labeled with near-infrared secondary antibodies (IRDyes 680 RD or 800 CW, LI-COR Biosciences, Lincoln, NE, USA), detected and quantified using Odyssey Fc Imaging System (LI-COR Biosciences). Primary antibodies were: phospho-PDH (ABS204, MERCK, Darmstadt, Germany), total-PDH (C54G1, cell signaling), antibeta ACTIN (A1978), and antialpha TUBULIN (T9026).

### 4.7. Lactate Assay

After treatment, the L-lactate assay was performed, according to the manufacturer’s instructions. Briefly, the supernatant of the cells was taken and introduced to the master reaction mix. Hereafter, an incubation period of 30 min took place at room temperature in the dark. Subsequently, the absorbance was measured at 570 nm, and the amount of L-lactate in the medium was calculated. Additionally, the cells were lysed, and the total amount of proteins was examined by the BCA protein assay (23227, Thermo Fisher); this step was performed to normalize the lactate values to the amount of proteins in the cell.

### 4.8. Seahorse Metabolic Profiling

Oxygen consumption rate (OCR) and extracellular acidification rate (ECAR) were determined using a Seahorse XF96 analyzer (Agilent Technologies, Santa Clara, CA, USA) as previously reported [100]. Briefly, 1.5 × 10^5^ cells were seeded in 96-well plates. Cells were then treated with DCA, either overnight (for OCR measurements) or for 3 h (during Seahorse run, for ECAR). Cells were equilibrated in unbuffered Dulbecco’s Modified Eagle Medium (DMEM) medium with 2 mM glutamine and 10 mM glucose at 37 °C in a CO_2_-free incubator and then measured using a Seahorse analyzer. To extract detailed information on the electron transport chain in mitochondria, specific inhibitors consisting of oligomycin, FCCP, rotenone and antimycin A were added sequentially. ECAR was normalized to the basal level.

### 4.9. ROS Production

The intracellular level of ROS was detected using 5-(6)-chloromethyl-2′,7′-dichlorodihydro-fluorescein diacetate (CM-H_2_DCFDA), an oxidation-sensitive fluorescent probe (Abcam, Cambridge, United Kingdom) as previously described [97]. Briefly, after treatment, cells were stained with 5 μM CM-H2DCFDA at 37 °C for 30 min. The mean fluorescence intensity was measured by a FACSCanto flow cytometer (BD Bioscience, Franklin lakes, NJ, USA) and analyzed by the Flowjo software (BD Bioscience).

### 4.10. NAD(P)H Measurement

The intracellular level of NAD(P)H was detected using the Cell Meter^TM^ intracellular NADH/NADPH flow cytometric analysis kit (AAT BioQuest, Sunnyvale, CA, USA), according to the manufacturer’s instructions. Briefly, after treatment, cells were stained with JZLA707 NAD(P)H sensor at 37 °C for 45 min. The mean fluorescence intensity was measured by a FACSCanto flow cytometer (BD Bioscience) and analyzed by the Flowjo software (BD Bioscience).

### 4.11. Radiation and Clonogenic Assay

After treatment, cells were irradiated at indicated doses on a 6 MV Linac (Varian Truebeam STx, Palo Alto, CA, USA; BrainLAB AG, Feldkirchen, Germany) and reseeded in 6-well plates for colony formation. Before seeding, cells were counted and normalized to control conditions. After 7–12 days, cultures were fixed with crystal violet and colonies (>50 cells) were counted. Survival fractions (SF) were fitted to the linear-quadratic model using GraphPad Prism 8 software (GraphPad Prism Software Inc, San Diego, CA, USA). Radiosensitization was evaluated at the level of 0.1 surviving fractions.

### 4.12. Three-dimensional (3D) Cell Cultures (Spheroids)

Spheroids were prepared with EMT6 and 4T1 cells by seeding 4000 cells/well in an ultra-low attachment 96-well plate (Corning, Corning, NY, USA). DCA was added to the medium when spheroids were around 500 µm in diameter. Afterwards, the spheroids were radiated at 8 Gy, and the treatment was washed with fresh medium that was refreshed every 3 days. Spheroid growth was monitored using the IncuCyte Live Cell Imaging System (Essen Bioscience) for 10 days.

### 4.13. Mouse Tumor Model

4T1 and EMT6 tumor cells (0.5 × 10^6^) were inoculated into the left hind limb of syngeneic Balb/c mice (female, 7–9 weeks old; Charles River Laboratories, L’Arbresle Cedex, France). When tumors reached approximately 150 mm^3^, mice were randomized and treated for 10 consecutive days with DCA 300 mg/kg (intraperitoneal or intratumoral). Mice were radiated with a single dose of 12 Gy (EMT6 tumors) or 15 Gy (4T1 tumors) or a fractionated radiation scheme of 5*4 Gy (EMT6 tumors) and 5*6 Gy (4T1 tumors) when tumors reached approximately 150 mm3. Radiation was delivered with a 6 MV Linac (Varian Truebeam STx). During the entire course of the experiment, tumors were measured with an electronic caliper, and the tumor volume was calculated using the formula: Volume = (Length × Width^2^) × 0.5. Experiments were approved by the Ethical Committee for use of laboratory animals of the Vrije Universiteit Brussel (ethical dossier numbers: 16-552-2 (18/4/2017) and 18-552-2 (1/6/2018)

### 4.14. Pimonidazole Staining on Tumor Sections

Tumors were inoculated as described in part 4.13. After the treatment, pimonidazole (60 mg/kg; Hypoxyprobe) was injected i.v. in the tail vein. Tumors were excised 1.5 h later, weighed, snap-frozen, and stored in plastic vials at –80 °C. Tumor sections (5 µm) were then immunostained by using an anti-Pimo rabbit antibody (Hypoxyprobe, Burlington, MA, USA), which was stained with an antirabbit FITC antibody (Abcam). The tumor slides were mounted with mounting fluid (DAKO mounting medium, Agilent) mixed with dapi (Sigma-Aldrich) and covered with a coverslip. Images were acquired with fluorescence confocal microscopy (EVOS FL, Thermo Fisher) and analyzed using ImageJ.

### 4.15. Statistics

All analyses were performed using GraphPad Prism 8.4.3. Data are expressed as mean ± SEM of at least three independent experiments unless otherwise indicated. Unpaired *t*-test, one-way ANOVA followed by a Dunnett’s multiple comparison test and two-way ANOVA with Dunnett’s, Sidak’s or Tukey’s multiple comparison test were used for statistical analyses: * *p* < 0.05, ** *p* < 0.01, *** *p* < 0.001, **** *p* < 0.0001.

## Figures and Tables

**Figure 1 ijms-21-09367-f001:**
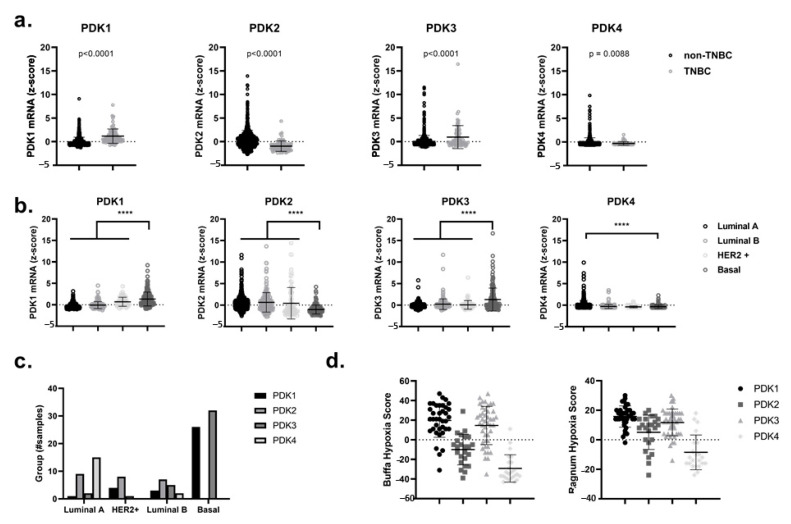
Upregulated mRNA expression for PDK1 and PDK3 correlates to hypoxia-related gene signatures in triple-negative and basal-like breast cancers. (**a**) PDK1–4 mRNA expression (RNA-seq) in non-triple-negative breast cancer (non-TNBC) subtype vs. TNBC cases. (**b**) PDK1–4 mRNA expression (RNA-seq) in different breast cancer subtypes classified by the Pam50 classification of the TCGA dataset (luminal A, luminal B, HER2+ and Basal). (**c**) Number of samples in TCGA dataset that have a *z*-score of *z* > 2 for PDK1–4 in different breast cancer subtypes classified by the Pam50 classification system. (**d**) The Buffa hypoxia score (left) and the Ragnum hypoxia score (right) correlates with the overexpression (*z* >2) of PDK1–4 mRNA. **** *p* < 0.0001.

**Figure 2 ijms-21-09367-f002:**
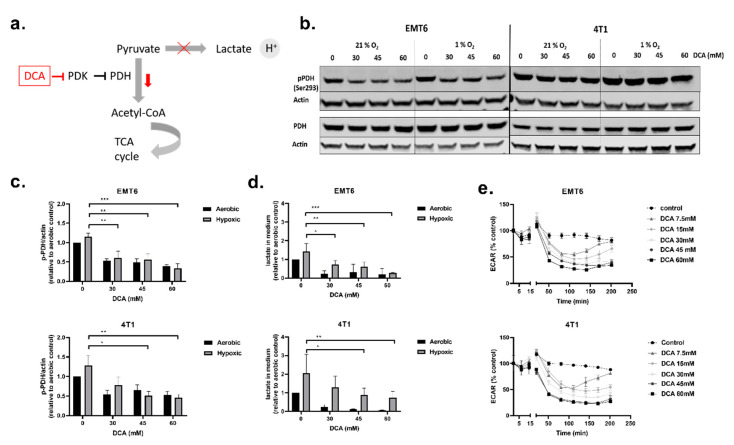
Dichloroacetate (DCA) decreases phosphorylated pyruvate dehydrogenase (PDH), lactate in the extracellular medium and ECAR of TNBC cells. (**a**) Schematic showing the influence of DCA on pyruvate dehydrogenase kinases (PDK) and downstream effects. (**b**) Representative Western blot of p-PDH (Ser293) and total PDH in 4T1 and EMT6 cells after treatment with DCA (30 mM, 45 mM and 60 mM) under aerobic and hypoxic conditions. (**c**) Normalized relative expression of p-PDH (Ser293) in 4T1 and EMT6 cells after treatment with DCA (30 mM, 45 mM and 60 mM) under aerobic and hypoxic conditions. (**d**) After treatment of DCA with indicated concentrations under aerobic or hypoxic conditions, lactate was measured with the L-lactate assay. (**e**) The ECAR of EMT6 and 4T1 cells was measured over time after the injection of indicated concentrations of DCA using the Seahorse analyzer. The extracellular acidification rate (ECAR) was expressed as the percentage relative to control. * *p* < 0.05, ** *p* < 0.01, *** *p* < 0.001.

**Figure 3 ijms-21-09367-f003:**
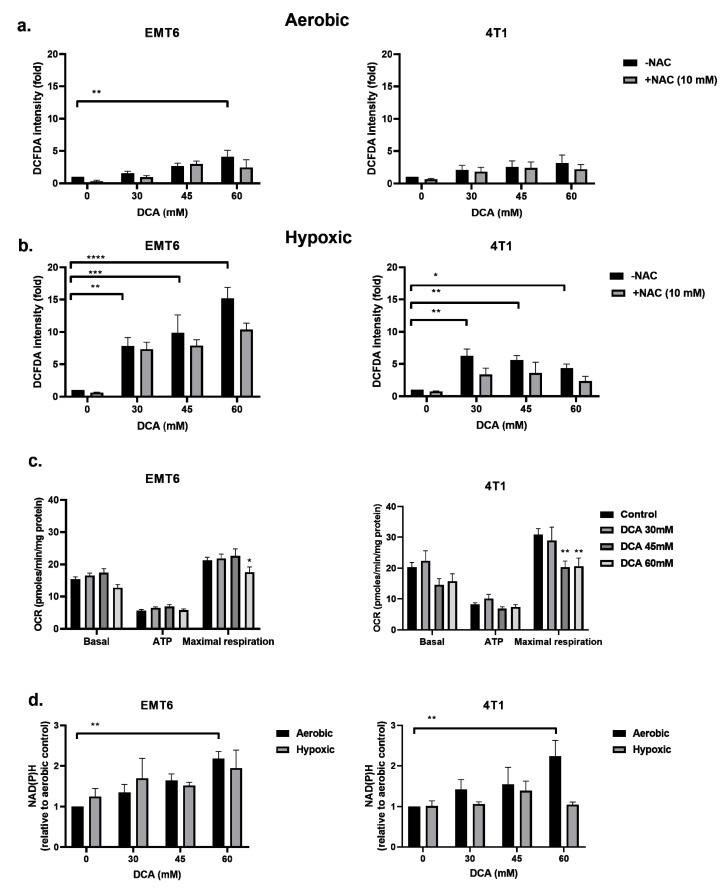
DCA induces ROS production in hypoxic cells by overloading the mitochondrial metabolism. EMT6 and 4T1 cells were treated with DCA overnight at indicated concentrations, and N-acetyl-cysteine (NAC) (10 mM) was added 1 h prior to and during treatment. ROS generation was measured by flow cytometry using CM-H2DCFDA probe under aerobic (**a**) and hypoxic (**b**) conditions. (**c**) Oxygen consumption rate (OCR) measurements were obtained over time by a Seahorse analyzer using the mitochondrial stress test. To extract detailed information on the electron transport chain in mitochondria, specific inhibitors consisting of oligomycin, FCCP, rotenone and antimycin A were added sequentially. Basal mitochondrial OCR, ATP-linked OCR and maximal OCR were calculated from these results. (**d**) NAD(P)H was measured by flow cytometry using a JZL1707 NAD(P)H sensor under aerobic and hypoxic conditions. * *p* < 0.05, ** *p* < 0.01, *** *p* < 0.001, **** *p* < 0.0001.

**Figure 4 ijms-21-09367-f004:**
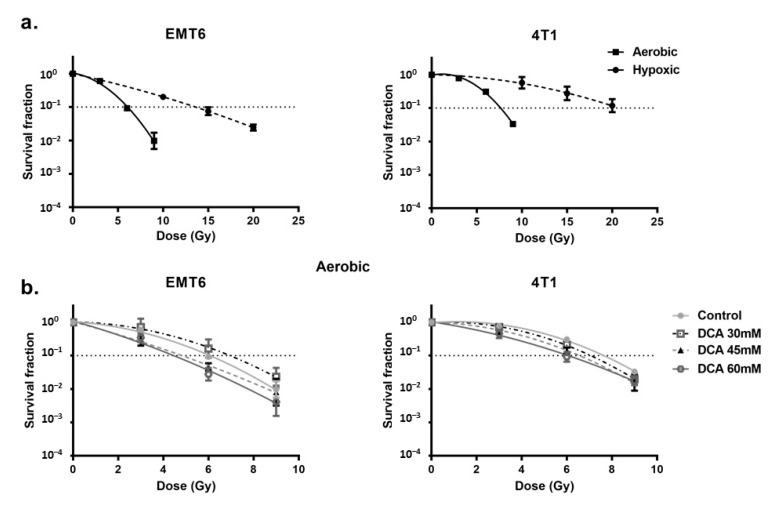
DCA has small radiosensitizing effects on aerobic tumor cells. EMT6 and 4T1 cells were treated with DCA overnight at indicated concentrations. (**a**) Radiosensitivity of EMT6 and 4T1 cells under either aerobic or hypoxic conditions. (**b**) Radiosensitizing effect of DCA under aerobic conditions.

**Figure 5 ijms-21-09367-f005:**
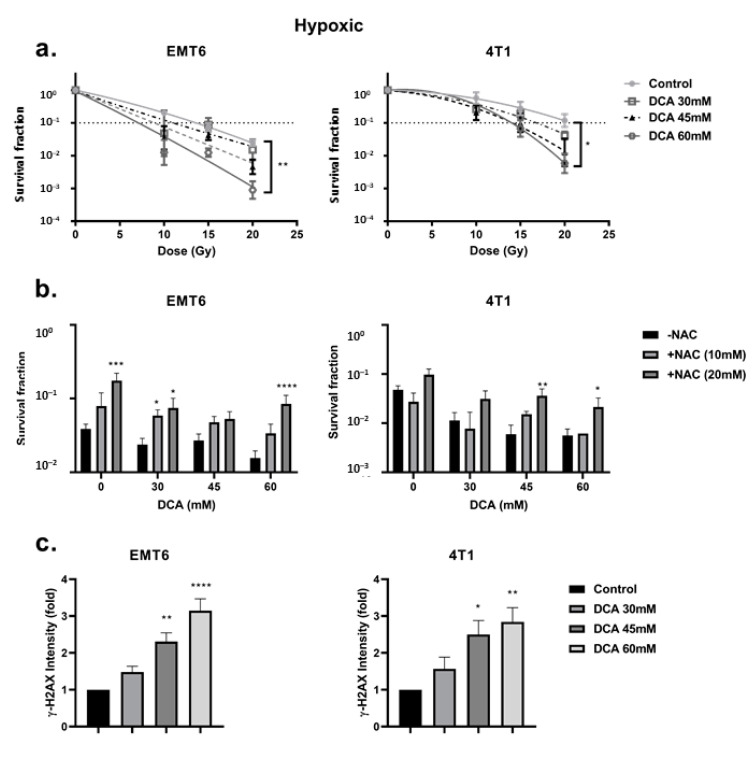
DCA radiosensitizes hypoxic tumor cells via the upregulation of ROS. EMT6 and 4T1 cells were treated with DCA overnight at indicated concentrations. (**a**) The radiosensitizing effect of DCA under hypoxic conditions assessed by colony formation assay. (**b**) Pretreatment of NAC (10 mM and 20 mM) reversed the radiosensitizing effect of different concentrations of DCA at 15 Gy for EMT6 and 20 Gy for 4T1. (**c**) Double-strand DNA breaks were analyzed by flow cytometry using gammaH2AX staining. * *p* < 0.05, ** *p* < 0.01, *** *p* < 0.001, **** *p* < 0.0001.

**Figure 6 ijms-21-09367-f006:**
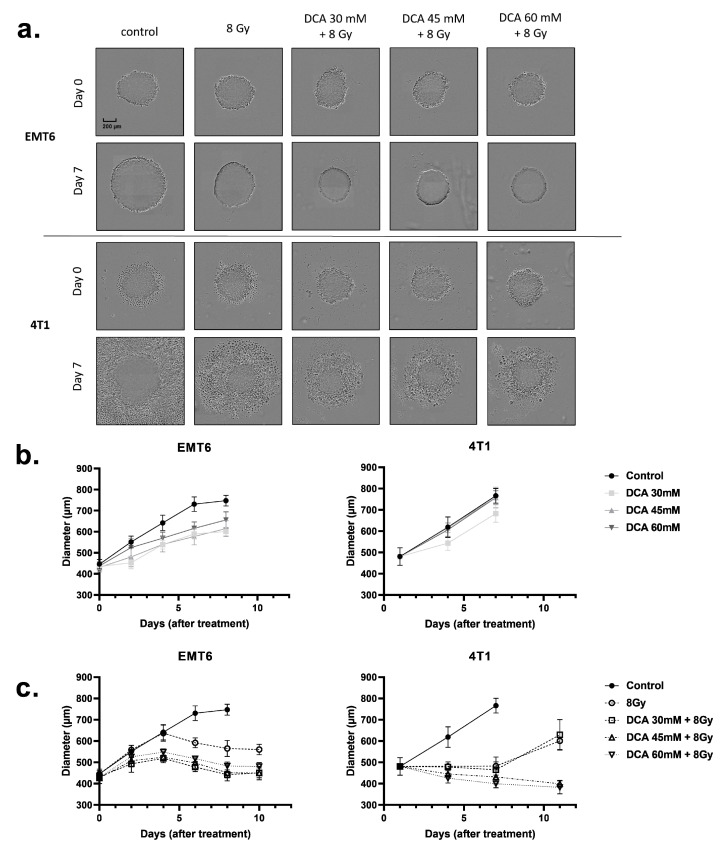
Treatment of DCA and 8 Gy radiation reduces the growth of EMT6 and 4T1 tumor spheroids. (**a**) Representative pictures and (**b,c**) time-dependent growth of EMT6 and 4T1 spheroids upon treatment with DCA at indicated concentrations and in combination with 8 Gy radiotherapy.

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
