# Peer review of "Dichloroacetate Radiosensitizes Hypoxic Breast Cancer Cells"

_ijms, 2020, doi:10.3390/ijms21249367_

Round 1
Reviewer 1 Report
Your main result is that DCA increases ROS in hypoxic cells, which leads to radiosensitivity. DCA target may be PDK 1and PDK3. However, the manuscript at this moment is likely to be premature. The following comments should be performed before being published.
Main points
- High concentration of DCA was required for your experiments. To specify the PDK involvement (to avoid non-specific effect of DCA), you should use si-PDK1 and si-PDK3.
- You showed DCA increased ROS levels. If DCA inhibits p-PDH levels, NADPH may be accumulated. NADPH may be used for superoxide formation through NADPH oxidase. To demonstrate this, you should use si-gp91 (a component of NADH oxidase), DPI and Apocynin (inhibitors of NADPH oxidase) to discriminate whether ROS is produced by NADPH oxidase. This experiment will clarify the mechanism by which ROS is produced by DCA.
Minor point
- Notes in font should be enlarged for readers. Fonts are too small to read easily, in particular in Fig. 2.
Author Response
To reviewer 1,
Manuscript ID: ijms-949096
We want to thank the reviewer for taking the time to revise our manuscript and his positive remarks and comments. We have addressed all issues below and in the manuscript. All your remarks are addressed point by point in the attached document and each change is outlined in track changes in the revised version of the manuscript.
Sincerely yours,
Mark De Ridder
E-mail: [email protected]

Reviewer 2 Report
The authors suggested that expressions of PDK1 and PDK3 in malignant breast cancers correlated with hypoxia, from big clinical data. On the basis, the authors proved in vitro an effectiveness of DCA, small molecule PDK inhibitor, on hypoxic breast cancer cell lines (EMT6 and 4T1) to upregulate ROS production and following radiosensitization, with sufficient experimentations. However, a failure for confirming the usability of DCA in vivo was regrettable. The authors have to write more about animal testing if they refer to the experiment in vivo.
I recommend to insert an explanation of Pam50 classification wherever.
The lines and/or symbols in Fig. 4b is wrong.
According to Fig. S1a, the cytotoxicity of DCA at 30-60 mM cannot be ignored. Therefore, I suggest using a dose modifying factor (DMF) instead of SF for the colony assay in Fig. 5, as it allows us to purely quantify the sensitizing effect of X-rays without the toxicity of DCA itself.
Descriptions about the animal experiments lacks a lot. The experimental condition and a declaration of compliance with laws and regulations must be included in the main article or the supplementary. Additionally, Body weight changes in mice should be shown to indicate that there are no toxic effects of DCA itself.
Author names in the reference list should be revised.
Author Response
To reviewer 2,
Manuscript ID: ijms-949096
We want to thank the reviewer for taking the time to revise our manuscript and his positive remarks and comments. We have addressed all issues below and in the manuscript. All your remarks are addressed point by point in the attached document and each change is outlined in track changes in the revised version of the manuscript.
Sincerely yours,
Mark De Ridder
E-mail: [email protected]

Round 2
Reviewer 1 Report
The authors did not do any experiments by using si-PDHK1,2,3, and 4.
In order to be published, the manuscript should ccontain the results of si-PDHK1, 2,3, and 4.
Author Response
To reviewer 1,
Manuscript ID: ijms-949096
We want to thank the reviewer for taking the time to revise our manuscript and his positive remarks and comments. We have addressed all issues below and in the manuscript. Your remark has been addressed in the attached document and each change is outlined in track changes in the revised version of the manuscript.
Sincerely yours,
Mark De Ridder
E-mail: [email protected]

Round 3
Reviewer 1 Report
The manuscript will be accepted.